# Evaluating conflict-of-interest governance among Canadian medical societies using a novel assessment tool: A cross-sectional study

Emily Rickard[1], Joel Lexchin [2]*

**1** Department of Social and Policy Sciences, University of Bath, Bath, United Kingdom, **2** School of Health Policy and Management, York University, Toronto, Ontario, Canada

* jlexchin@yorku.ca

## Abstract

### Background

Medical societies shape clinical standards, develop practice guidelines, and provide continuing education to physicians. Many receive funding from pharmaceutical and medical device companies, creating conflicts of interest (COI); however, organizational-level COI governance in medical societies remains largely unexamined. We developed what is, to our knowledge, the first comprehensive tool to assess medical society COI policies and applied it to Canadian medical societies.

### Methods and findings

We identified 68 Canadian medical societies meeting our inclusion criteria and systematically searched the publicly accessible sections of their websites for COI policies. Two researchers independently analyzed policies using our 51-item tool covering nine domains: continuing medical education/continuing professional development (CME/CPD), leadership COI, clinical practice guidelines, industry relationships, research funding, external funding, annual general meetings, staff COI, and society journals. Only 33 societies (48.5%) had any publicly available COI policy, and those with policies addressed an average of only 1.85 domains (20.5% of possible domain coverage). The most frequently addressed domain was CME/CPD (22 societies, 32.4%), followed by leadership COI (12 societies, 17.6%). Within domains where policies existed, societies covered an average of only 32.5% of relevant items. The most common policy item was speakers' declaration of COI in CME/CPD activities (16 societies, 23.5%). Policies addressing research funding, external funding, annual meetings, and staff COI were rare (1.5–5.9% of societies). Where present, policies were generally restrictive but narrow in scope.

**Data availability statement:** All relevant data are within the manuscript and its Supporting Information files.

**Funding:** This study was financially supported by the United Kingdom's Economic and Social Research Council in the form of a postdoctoral fellowship awarded to ER (UKRI2500). No additional external funding was received for this study. The funder had no role in study design, data collection and analysis, decision to publish, or preparation of the manuscript.

**Competing interests:** Emily Rickard declares no conflict of interest Between 2022-2025, Joel Lexchin received payments for writing a brief for a legal firm on the role of promotion in generating prescriptions for opioids, for being on a panel about pharmacare and for co-writing an article for a peer-reviewed medical journal on semaglutide. He is a member of the Boards of Canadian Doctors for Medicare and the Canadian Health Coalition. He receives royalties from University of Toronto Press and James Lorimer & Co. Ltd. for books he has written. He has received funding from the Canadian Institutes of Health Research in the past.

## Conclusions

Most Canadian medical societies lack comprehensive COI governance, and existing policies are fragmented and reactive. More transparent, comprehensive, and enforceable frameworks are needed to protect institutional independence and public trust. The assessment tool may be adaptable for use in other jurisdictions to support international standardization and best-practice development.

## Introduction

Medical societies and associations are voluntary membership organizations of physicians who share a common expertise in a medical specialty (e.g., respirology) or a common interest in a particular area of practice (e.g., environmental protection). Societies serve important purposes; they provide continuing professional education to their membership through courses and annual conferences, they advocate to government and others on behalf of their membership and the patients that they treat, and they promote continual improvement in their area of knowledge.

While some of the revenue necessary to undertake their activities comes from annual membership dues, they also frequently receive grants, donations and fees from activities such as renting booths at meeting and conventions to commercial entities, primarily pharmaceutical and medical device companies (henceforth companies) and payments directly from companies. Among 65 Canadian medical societies, twenty-three (35.4%) accepted pharmaceutical company sponsorship for their activities and 25% reported sponsorship for their annual general meetings. At the same time, only 10 of the societies had policies about managing their interactions with companies [1]. Concerns have been raised about whether the financial conflicts of interest (COI) of societies have biased their views about a range of issues including the acceptability of generic immunosuppressive drugs in solid organ transplantation [2] and the inclusion of the COI of authors in clinical practice guidelines about hormone replacement therapy [3].

To date, there has only been a single study of policies covering a limited number of domains: the disclosure of COI, management of such disclosures or failure to disclose, and provisions on acceptance of external funding for the organization's activities of 17 United Kingdom health professional organizations (organizations representing healthcare professionals who have medical or professional licenses, certificates or diplomas) [4]. The aim of this study was to develop a tool for analyzing medical society COI policies and then to apply that tool to the policies of Canadian medical societies.

## Methods

### List of medical societies

We defined a medical society as an organization with voluntary membership of physicians structured around either a specific specialty or a specific area of interest. Therefore, organizations that had mandatory membership (e.g., provincial licensing

colleges), were open to all physicians (e.g., provincial medical associations), that accredited physicians (e.g., College of Family Physicians of Canada, Royal College of Physicians and Surgeons of Canada) or were open to non-physicians (e.g., Nova Scotia paramedic society) were excluded.

In the absence of any single consolidated list of medical societies, a list was constructed based on three sources: medical associations based in Canada [5], affiliates of the Canadian Medical Association [6] and national specialty societies listed by the Royal College of Physicians and Surgeons [7].

## Medical society policies

Policies were identified through a systematic search of publicly accessible sections of each society's website. We focused on publicly available policies because these represent the standards communicated to external stakeholders and therefore constitute the most transparent and publicly accountable form of organizational governance. Restricting the search to publicly accessible materials also ensured methodological reproducibility, as access to password-protected or internal documents varies across organizations and investigators. We did not contact medical societies to request additional policies because our objective was to evaluate policies accessible to any interested party. Consequently, our analysis reflects the policies that societies make publicly available rather than any internal policies that may exist.

COI policies were defined as policies that covered relationships with commercial entities that provided funding or other forms of sponsorship to the society. Only policies governing the activities of societies or people representing the society were downloaded. We also searched for whether societies owned journals or if the societies had an official journal on the grounds that if such a journal existed it should be governed by a COI policy. If a policy governing journal COI was on the society's main website or had a hyperlink from the main website, then we included the policy in our analysis. If the policy was only on a separate journal website it was excluded. Policies governing the independent activities of individual society members were excluded.

## Tool to analyze policies

No comprehensive tool for analyzing organizational COI policies currently exists. In order to create one, we developed a structured assessment tool through a multi-step process grounded in existing literature and expert consensus. First, we conducted a targeted review of published research to identify items that should be included in COI policies [4,8–13], which we then synthesised into an initial item pool. Second, items were organised into thematic domains representing major areas where COIs have been shown to influence medical society activities, specifically clinical practice guielines [14], society leadership [15], continuing medical education [16], industry-sponsored research [17], external funding [18], annual meetings [19], society journals [20], and policies governing relationships with industry [21]. Third, the draft tool underwent iterative review by 10 researchers with extensive experience in developing and analyzing COI policies. The group evaluated item relevance, domain structure, clarity, and completeness. Feedback informed a revised version, which was recirculated for final comments. The final tool consisted of discrete *policy items* (i.e., specific policy provisions or rules) grouped into broader thematic *domains* (i.e., categories of related policy items addressing a common policy area). Items were equally weighted to avoid imposing subjective judgements regarding the relative importance of policy components and to maintain replicability of scoring. This development process aligns with established approaches used in prior institutional COI policy assessment tools, for example academic medical center policies [22]. The final tool is provided in S1 File.

## Analysis of policies

Policies were independently read and scored by two researchers. Disagreements were resolved by consensus. Each item in the tool was scored from 0 to 2, where 0 represented the most restrictive interpretation (e.g., the activity was not

allowed), 1 represented a partially restrictive position, and 2 represented the most permissive interpretation (e.g., no restrictions).

Breadth of coverage was defined as the number of domains for which a society had at least one policy item. Depth of coverage was defined as the proportion of items addressed within a domain among societies that had a policy in that domain. Item-level coverage was calculated as the number and percentage of societies addressing each specific item.

For each individual domain we reported the number of societies that covered at least one item in the domain, e.g., Annual General Meeting – 10 societies, Clinical Practice Guidelines – 15 societies, etc. We counted and reported the number of domains covered by each society, e.g., 5 societies covered one domain, 20 societies covered two domains, etc. We calculated and reported the mean number of items covered by all societies for each individual domain and the mean score (out of 2). Finally, for each society we calculated (1) the absolute score, defined as the sum of all item scores; (2) the percentage of the maximum possible score, calculated as the absolute score divided by the maximum possible score if all applicable items were scored "2", e.g., if a society's policy mentioned 15 different items then the maximum score would be 30 and we would report an absolute score of 20 and a percent score of 66%; and (3) the mean overall score, calculated as the average item score (0–2).

### Ethics and patient involvement

No patients were involved in this study. All data were publicly available and ethics approval was not necessary. There was no specific funding for the study. The 2007 STROBE guidelines for cross-sectional studies were followed [23].

## Results

The final tool covered 9 different domains and 51 items, 3–11 items per domain (Table 1). After eliminating duplicates and organizations that did not meet our definition of a medical society, 68 societies were identified (Fig 1) and their websites were initially searched between March 29–30, 2025 and again between May 10-June 10, 2025. (S2 File has a full list of the medical societies and the relevant websites.) Of these, only 33 societies (48.5%) had at least one publicly available policy.

### Policy presence and breadth

Over half of the societies (35/68, 51.5%) lacked policies addressing any of the nine assessed domains (Table 2). Among the 33 societies with at least one policy, breadth of coverage ranged from 1 to 5 domains (11% to 56% of possible coverage), with an average of 1.85 domains addressed (SD = 1.42), equivalent to 20.5% of the nine domains. The average

Table 1. Number of domains and items per domain.

| Name of domain | Items in domain – n |
| --- | --- |
| Annual general meeting | 3 |
| Clinical practice guidelines | 9 |
| Conflict of interest of society leadership, including board | 4 |
| Conflict of interest of society staff | 4 |
| Continuing medical education/continuing professional development | 11 |
| External funding to society | 3 |
| Policy on relationship with companies | 8 |
| Research funding | 5 |
| Society journal | 4 |

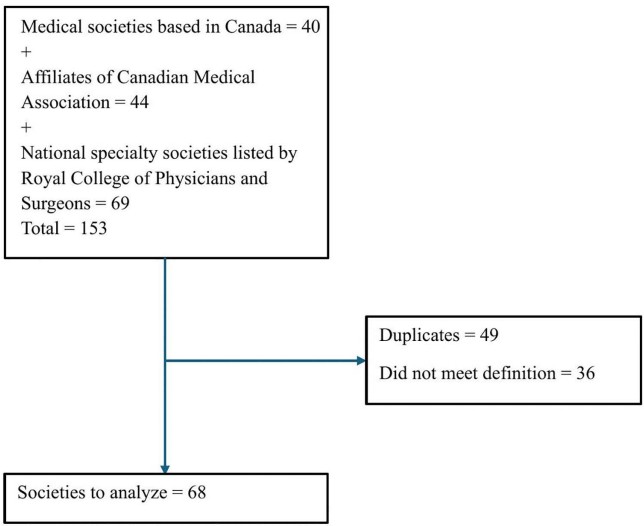

**Fig 1. Identification of societies.**

**Table 2. Number of domains with a policy covering at least one item in a domain.**

| Domains addressed – n | Societies – n (% of all societies) |
|---|---|
| 0 | 35 (51.47) |
| 1 | 22 (32.35) |
| 2 | 4 (5.88) |
| 3 | 0 |
| 4 | 3 (5.88) |
| 5 | 3 (4.41) |

number of domains covered across all 68 societies was 0.90 (SD = 1.35), corresponding to 10% of the nine possible domains.

The most frequently addressed domain was continuing medical education/continuing professional development (CME/CPD), with 22 societies (32.4%) publishing a relevant policy (Table 3). Policies relating to conflict of interest (COI) of society leadership including the board were the second most common (17.6%), followed by clinical practice guidelines (11.8%) and policies on the relationship with industry (10.3%). Very few societies addressed research funding (5.9%), external funding (4.4%), and only one society (1.5%) had a policy covering annual general meetings or COI of society staff. Three (7.7%) of the 39 societies with a journal published a relevant policy on their website.

## Depth of coverage within policy domains

Where societies had policies, depth of coverage was limited – on average, societies covered 32.5% of items within the domains for which they had a policy. Average item coverage across all policies ranged from 21.4% (relationship with industry) to 50% (COI of society staff, although this was based on 1 policy only) (Table 4). Within the most frequently addressed domain (CME/CPD), depth was low, with an average of 25.6% of items addressed. In contrast, the second most frequent policy (COI of society leadership including the board) had higher coverage (45.8%).

**Table 3. Breadth of coverage across domains.**

| Policy domain | Societies with a policy – n (% of 68 societies) |
|---|---|
| Continuing medical education/continuing professional development | 22 (32.4) |
| COI of society leadership including board | 12 (17.6%) |
| Clinical practice guidelines | 8 (11.8%) |
| Policy on relationship with industry | 7 (10.3%) |
| Research funding | 4 (5.9%) |
| External funding to society | 3 (4.4%) |
| Society journal (if exists)* | 3 (7.7%) |
| Annual general meeting | 1 (1.5%) |
| COI of society staff | 1 (1.5%) |

*percent is of the 39 societies with a journal.

**Table 4. Depth of coverage within policy domains.**

| Policy domain | Total items | Maximum percent of items covered | Mean percent of items covered | Mean items covered |
|---|---|---|---|---|
| Continuing medical education/continuing professional development | 11 | 63.6% | 25.6% | 2.82 |
| COI of society leadership including board | 4 | 50% | 45.8% | 1.83 |
| Clinical practice guidelines | 9 | 44.4% | 31.9% | 2.88 |
| Policy on relationship with industry | 8 | 25% | 21.4% | 1.71 |
| Research funding | 5 | 80% | 45% | 2.25 |
| External funding to society | 3 | 66.7% | 44.4% | 1.33 |
| Society journal (if exists)* | 4 | 75% | 41.7% | 1.67 |
| Annual general meeting | 3 | 33.3% | 33.3% | 1 |
| COI of society staff | 4 | 50% | 50% | 2 |

*percent is of the 39 societies with a journal.

The most frequently addressed item was *Speakers' declaration of conflict of interest* within CME/CPD, covered by 16 societies (23.5%) (Table 5). This was followed by *Current COI with industry/company allowed* (12 societies, 17.6%) under COI of society leadership. Two items were addressed by 10 societies each: *Current COI with industry/company disclosure policy* (leadership COI) and *Adherence to the Royal College of Physicians and Surgeons of Canada National Standard* (CME/CPD). No other individual item was addressed by more than 10 societies.

### Policy strictness and score distribution

Policies were generally restrictive when present, with variability across domains (Table 6). While policies for *COI of society leadership* were the most permissive (domain mean 1.09), policies covering *CME/CPD* (0.31) and *clinical practice guidelines* (0.43) were comparably restrictive. The few policies that addressed research funding and society journals were consistently highly restrictive (0) on the items they addressed.

At the society level, most policy-holding societies had a mean overall score clustered at 0 to <0.5 (19 societies) (predominantly restrictive), with a smaller subset (7 societies) adopting moderately restrictive positions (0.5 to <1.0), and seven societies were comparably permissive (mean scores ≥1.0).

**Table 5. Items addressed most frequently by societies.**

| Item | Domain | Societies – n (%)* |
|---|---|---|
| Speakers' declaration of COI | CME/CPD | 16 (23.5%) |
| Current COI with industry/company allowed | COI Leadership | 12 (17.6%) |
| Current COI with industry/company disclosure policy | COI Leadership | 10 (14.7%) |
| Adherence to RCPSC National Standard | CME/CPD | 10 (14.7%) |
| Independent development of content by society | CME/CPD | 9 (13.2%) |
| Independent development of domain by society | CME/CPD | 8 (11.8%) |
| Company funding for meeting | CME/CPD | 7 (10.3%) |
| Use of society name and logo by company | Relationship with Industry | 6 (8.8%) |
| Individuals declare COI before selection | Clinical Practice Guidelines | 6 (8.8%) |

*Percentage is of all 68 societies.

**Table 6. Average scores within domains.**

| Policy domain | Policies – n | Mean score* | SD |
|---|---|---|---|
| Annual general meeting | 1 | 1 | n/a |
| Clinical practice guidelines | 8 | 0.43 | 0.51 |
| COI of society leadership including board | 12 | 1.09 | 0.29 |
| COI of society staff | 1 | 1 | 0 |
| Continuing medical education/continuing professional development | 22 | 0.31 | 0.56 |
| External funding to society | 3 | 0.75 | 0.5 |
| Policy on relationship with industry | 7 | 0.67 | 0.49 |
| Research funding | 4 | 0 | 0 |
| Society journal (if exists) | 3 | 0 | 0 |

*Mean calculated on scores for all items per domain.

## Society-level variation in COI policy coverage and strength

Coverage and policy restrictiveness varied substantially across societies (Table 7), ranging from 1 to 12 items (median: 2; IQR: 2–8). While the Canadian Pain Society addressed 12 items across 5 domains, most societies (21/33, 63.6%) covered ≤3 items, indicating limited policy scope even among those with publicly available policies. Only six societies (18.2%) addressed ≥8 items. Coverage patterns varied not only in scope but in approach. Some societies demonstrated depth within limited domains (e.g., Canadian Association of Emergency Physicians: 7 items in 1 domain), while others adopted broader but shallower coverage (e.g., Canadian Rheumatology Association: 8 items across 4 domains).

Policy restrictiveness also varied considerably. Absolute scores ranged from 0 (highly restrictive policies) to 7 (more permissive policies), with corresponding percentages of maximum permissiveness ranging from 0% to 75%. The majority of societies (22/33, 66.7%) scored below 30% of maximum permissiveness, indicating predominantly restrictive approaches. However, seven societies scored ≥50% of maximum, reflecting substantially more permissive stances in COI policies. Eleven societies adopted fully restrictive positions across all addressed items (0% of maximum), whereas a smaller number (e.g., Canadian Psychiatric Association) demonstrated comparatively permissive approaches (up to 75% of maximum).

**Table 7. Society-level coverage and restrictiveness of COI policies.**

| Medical society | Domains covered – n | Items covered – n | Mean score | Absolute score | Max possible score | % of Max* |
|---|---|---|---|---|---|---|
| Canadian Pain Society | 5 | 12 | 0.5 | 6 | 24 | 25 |
| Association of Medical Microbiology and Infectious Disease Canada | 5 | 11 | 0.45 | 5 | 22 | 22.7 |
| Canadian Association for the Study of Liver | 2 | 11 | 0.45 | 5 | 22 | 22.7 |
| Canadian Urological Association | 4 | 10 | 0.3 | 3 | 20 | 15 |
| Canadian Association of Radiologists | 4 | 9 | 0.77 | 7 | 18 | 38.9 |
| Canadian Thoracic Society | 5 | 9 | 0.67 | 6 | 18 | 33.3 |
| Canadian Rheumatology Association | 4 | 8 | 0.75 | 6 | 16 | 37.5 |
| Canadian Paediatric Society | 4 | 8 | 0.5 | 4 | 16 | 25 |
| Canadian Association of Emergency Physicians | 1 | 7 | 0.57 | 4 | 14 | 28.6 |
| Canadian Cardiovascular Society | 2 | 6 | 0.33 | 2 | 12 | 16.7 |
| Canadian Society of Internal Medicine | 1 | 5 | 0.2 | 1 | 10 | 10 |
| Canadian Orthopaedic Association | 2 | 4 | 0.25 | 1 | 8 | 12.5 |
| Canadian Society of Nephrology | 2 | 3 | 0.67 | 2 | 6 | 33.3 |
| Canadian Association of Gastroenterology | 1 | 3 | 0.33 | 1 | 6 | 16.7 |
| Society of Gynecologic Oncology of Canada | 1 | 3 | 0.33 | 1 | 6 | 16.7 |
| Canadian Society for Clinical Investigation | 1 | 3 | 0 | 0 | 6 | 0 |
| Canadian Psychiatric Association | 1 | 2 | 1.5 | 3 | 4 | 75 |
| Canadian Academy of Psychiatry and the Law | 1 | 2 | 1 | 2 | 4 | 50 |
| Canadian Anesthesiologists' Society | 1 | 2 | 1 | 2 | 4 | 50 |
| Canadian Society for Transfusion Medicine | 1 | 2 | 1 | 2 | 4 | 50 |
| Canadian Society of Addiction Medicine | 1 | 2 | 1 | 2 | 4 | 50 |
| Canadian Academy of Geriatric Psychiatry | 1 | 2 | 0 | 0 | 4 | 0 |
| Canadian Association of Medical Oncologists | 1 | 2 | 0 | 0 | 4 | 0 |
| Canadian Critical Care Society | 1 | 2 | 0 | 0 | 4 | 0 |
| Canadian Dermatology Association | 1 | 2 | 0 | 0 | 4 | 0 |
| Canadian Ophthalmological Society | 1 | 2 | 0 | 0 | 4 | 0 |
| Canadian Society of Endocrinology & Metabolism | 1 | 2 | 0 | 0 | 4 | 0 |
| Canadian Association of Interventional Cardiology | 1 | 1 | 1 | 1 | 2 | 50 |
| Trauma Association of Canada | 1 | 1 | 1 | 1 | 2 | 50 |
| Canadian Academy of Child and Adolescent Psychiatry | 1 | 1 | 0 | 0 | 2 | 0 |
| Canadian Association of Neuropathologists | 1 | 1 | 0 | 0 | 2 | 0 |
| Canadian Society of Allergy and Clinical Immunology | 1 | 1 | 0 | 0 | 2 | 0 |
| Cell Therapy Transplant Canada | 1 | 1 | 0 | 0 | 2 | 0 |

*Percentage of maximum represents the absolute score as a percentage of the maximum (most permissive) possible score if all items were scored as 2. Lower percentages indicate more restrictive overall policy positions, whereas higher percentages indicate more permissive policies.

## Discussion

This study provides one of the first systematic examinations of conflict of interest (COI) policies among Canadian medical societies and introduces a novel, evidence-informed, expert-reviewed tool for assessing organisational COI governance. Although developed within the Canadian context, the tool's structure and scoring approach are adaptable to other jurisdictions and regulatory environments. By establishing such a framework, this study lays the groundwork for international comparative analyses of institutional COI governance.

Our findings reveal that COI governance among Canadian medical societies is frequently absent, and when present, tends to be narrow in scope, disclosure-focused, fragmented, and insufficiently institutionalized. Fewer than half of societies had any publicly available COI policy, and those that did typically addressed a single domain – most commonly CME/CPD, where speakers' declaration of COI emerged as the most frequently cited item across all domains. Policies governing domains such as research funding, external funding, annual general meetings, staff COIs, and journals were rare. Even within policies, depth of coverage was modest: on average, societies addressed fewer than one-third of the relevant items within each domain. While this could be interpreted as a policy weakness, it may also reflect the comprehensiveness of our tool.

Our findings also indicate that policy coverage is more frequently observed in domains subject to external accreditation or scrutiny. For example, many CME/CPD policies explicitly referenced the Royal College of Physicians and Surgeons of Canada National Standard. This alignment is consistent with COI governance being structured around meeting external requirements rather than embedding institution-wide preventative safeguards. Additionally, policies more frequently included disclosure requirements than structural safeguards such as divestment or recusal. This pattern may suggest a comparatively reactive rather than proactive approach to managing conflicts of interest, which may have implications for institutional independence. This could, in turn, have clinical implications: when societies with limited COI governance develop practice guidelines or accredit educational programs, they risk introducing commercial bias into clinical decision-making, potentially affecting treatment recommendations and patient care.

Beyond individual clinical decisions, weak COI governance may pose risks to institutional credibility and public trust. In the absence of structured and transparent COI frameworks, Canadian medical societies may be vulnerable to both real and perceived bias, including in the context of industry-sponsored CME/CPD [16] and guideline development [24], where commercial influence has been well-documented. Such perceived conflicts may undermine the professional credibility and the profession's standing as an independent voice for medical evidence and patient interests.

## Comparison with international evidence

International evidence reveals similar challenges. In Italy, Fabbri et al. found that 65% of societies received industry sponsorship for their last conference, yet only 4.6% published an ethical code and 6.1% disclosed annual financial reports, with sponsorship levels not reduced by the presence of such policies [25]. Vercellini et al. [26] found that most Italian gynaecological societies provided CME activites and displayed industry advertising on their website, yet none had publicly available COI policies or financial disclosures. In Australia, Kerridge et al. [27] reported that 20 of 29 responding organizations (from 63 contacted) had policies or guidelines governing industry relationships. However, these policies appeared limited in scope: 62% of organizations did not require speakers to declare COIs at educational meetings. This contrasts with our findings, where speaker declaration requirements was the most frequently addressed policy item across all items in our study. The increase in the percent of policies requiring speaker disclosure may reflect an expanding recognition of the influence of COI. Globally, Brems et al. [8] found that while most guideline-producing bodies had a relevant COI policy, only 3% met all Institute of Medicine standards and violations of internal policies were common. These studies collectively demonstrate that policy presence alone does not equate to effective COI management.

## Limitations

Due to the lack of a single consolidated list of Canadian medical societies, some may have been missed. The websites of most societies have password-protected pages and additional policies may have been present on these pages. The tool that was developed has face value and underwent expert review, but it has not been previously used and may have biases that reflect the point of view of the group that developed it. The choice of domains and the items included under each domain was based on what the individuals developing the tool felt were important, but the tool could have included additional domains and items. Our findings do not extend to organizations excluded under our eligibility criteria (e.g.,

regulatory colleges, accrediting bodies, broad physician associations, or multidisciplinary organizations) or to medical societies in other countries.

## Conclusion

Canadian medical societies show inadequate COI governance, characterised by low policy coverage, limited depth, and variable levels of restrictiveness. Our findings highlight a need for comprehensive, enforceable, and transparent COI governance frameworks. The assessment tool introduced here offers a foundation for evaluating and comparing such policies internationally. Future research should extend this analysis to other jurisdictions, assess policy implementation and enforcement, and examine its impact on professional integrity and public trust.

## Supporting information

**S1 File. Medical societies included and URL link to conflict-of-interest policies.**
(XLSX)

**S2 File. Scoring tool domains and topics and explanation of score assigned.**
(DOCX)

## Acknowledgments

Drs. Sharon Batt, Alice Fabbri, Adriane Fugh-Berman, Marc-André Gagnon, Lisa Parker and Sheryl Spithoff all contributed to developing the scoring tool.

## Author contributions

**Conceptualization:** Joel Lexchin.

**Data curation:** Emily Rickard, Joel Lexchin.

**Formal analysis:** Emily Rickard, Joel Lexchin.

**Investigation:** Emily Rickard, Joel Lexchin.

**Methodology:** Emily Rickard, Joel Lexchin.

**Writing – original draft:** Emily Rickard, Joel Lexchin.

**Writing – review & editing:** Emily Rickard, Joel Lexchin.

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
