## [Decision Letter · Decision Letter 0]

12 Feb 2026

PONE-D-25-63289Evaluating conflict-of-interest governance among Canadian medical societies using a novel assessment tool: a cross-sectional studyPLOS One

Dear Dr. Lexchin,

Thank you for submitting your manuscript to PLOS ONE. After careful consideration, we feel that it has merit but does not fully meet PLOS ONE’s publication criteria as it currently stands. Therefore, we invite you to submit a revised version of the manuscript that addresses the points raised during the review process.

We look forward to receiving your revised manuscript.

Kind regards,

Tope Michael Ipinnimo, MBBS, MPH, FWACP, FMCPH

Academic Editor

PLOS One

Journal Requirements:

Additional Editor Comments:

Kindly address the comments of the reviewers.

Reviewers' comments:

Reviewer's Responses to Questions

**Comments to the Author**

1. Is the manuscript technically sound, and do the data support the conclusions?

Reviewer #1: Yes

Reviewer #2: Yes

2. Has the statistical analysis been performed appropriately and rigorously?

Reviewer #1: Yes

Reviewer #2: Yes

3. Have the authors made all data underlying the findings in their manuscript fully available?

Reviewer #1: Yes

Reviewer #2: Yes

4. Is the manuscript presented in an intelligible fashion and written in standard English?

Reviewer #1: No

Reviewer #2: Yes

5. Review Comments to the Author

Reviewer #1: The manuscript is well written and organized, and the data availability and reporting standards meet PLOS ONE requirements. I have no major concerns except from some minor issues; Typo/wording: “Policy presense” in the Results section should be “Policy presence.”

- Consistency and clarity: Consider defining early what counts as a “policy item” vs. a “domain,” then keep the wording consistent throughout.

Reviewer #2: The manuscript presents a novel, systematic evaluation of conflict-of-interest (COI) policies among Canadian medical societies. The topic is of high relevance to medical ethics, professional self-regulation, and public trust. The study is well-written, well-structured, the methodology is clearly described, and the findings are stark and impactful, clearly demonstrating a major gap in transparent COI governance. However, several major methodological and interpretive issues need to be addressed to strengthen the validity of the conclusions and the utility of the tool, these include:

-) MAJOR COMMENTS:

1. The core of this study is the novel 51-item tool. While the expert review process is noted, the manuscript does not thoroughly address the tool's construct validity in terms of system scoring, Item and Domain Selection, and weighing.

2. It is crucial to have a robust justification for the decision and its ramifications, as limiting the search to public websites represents a major methodological choice.

3. Tables 3, 4, and 6 are dense and somewhat confusing. Table 3's "Depth of coverage" columns are hard to parse. Consider simplifying. For Table 6, the "% of Max" column is useful but requires careful reading. Ensure the caption/footnote clearly explains that a lower percentage indicates a more restrictive overall policy stance.

4. The low "depth" scores are highlighted. This could be interpreted not only as a policy weakness but also that the tool is very comprehensive. This dual interpretation should be acknowledged.

5. The discussion suggests policies are "reactive" and "compliance-driven." While plausible, this is still a speculative from a cross-sectional study. It is thus strongly suggested to refine informal phrasing (such as “implying” or “indicating”) to underscore the relational elements within your observations.

-----------

-) MINOR COMMENTS:

1. In the abstract, the authors wrote "Systematically searched their websites".

Could be slightly more specific (e.g., "publicly accessible sections of their websites").

2. Line 38: Typo: "a 38 nd staff COI" -> "and staff COI".

3. Line 288: "The analysis of the policies does not apply to policies of excluded Canadian organizations..." This sentence is awkward. Rephrase for clarity.

3. In Discussion section (Subtle: International Comparison): The comparison with the Australian study (Kerridge et al. 2005) is excellent as it contrasts your finding on speaker declarations. Emphasize this point of difference more. .

6. PLOS authors have the option to publish the peer review history of their article (what does this mean?). If published, this will include your full peer review and any attached files.

Reviewer #1: **Yes:** Ayokunle Oluwadoyinsola Adedipe

Reviewer #2: No

---

## [Author Response · Author response to Decision Letter 1]

16 Mar 2026

Reviewer #1:

Reviewer #1 comments

The manuscript is well written and organized, and the data availability and reporting standards meet PLOS ONE requirements. I have no major concerns except from some minor issues

Authors’ response

Thank you for your positive feedback upon your reading of our manuscript.

Reviewer #1 comments

Typo/wording: “Policy presense” in the Results section should be “Policy presence.”

Authors’ response

Thank you for spotting this typo, we have now corrected the spelling.

Reviewer #1 comments

Consistency and clarity: Consider defining early what counts as a “policy item” vs. a “domain,” then keep the wording consistent throughout.

Authors’ response

Thank you for flagging that this distinction was not made clear from the outset. We have now revised the paragraph in the methods section under ‘Tool to analyze policies’ to clarify how these terms are interpreted. Specifically, we now say “The resulting tool consisted of discrete policy items (i.e. specific policy provisions or rules) grouped into broader thematic domains (i.e. categories of related policy items addressing a common policy area.”

Reviewer #2 comments

The manuscript presents a novel, systematic evaluation of conflict-of-interest (COI) policies among Canadian medical societies. The topic is of high relevance to medical ethics, professional self-regulation, and public trust. The study is well-written, well-structured, the methodology is clearly described, and the findings are stark and impactful, clearly demonstrating a major gap in transparent COI governance. However, several major methodological and interpretive issues need to be addressed to strengthen the validity of the conclusions and the utility of the tool, these include

Authors’ response

We are very grateful for your time in engaging with our manuscript. We have enjoyed engaging with your suggestions and feel the manuscript has been strengthened as a result.

Reviewer #2 comments

Major comments

1. The core of this study is the novel 51-item tool. While the expert review process is noted, the manuscript does not thoroughly address the tool's construct validity in terms of system scoring, Item and Domain Selection, and weighing.

Authors’ response

Thank you for your suggestion to expand our discussion around the tool’s construct validity. In response, we have expanded the Methods section to clarify the conceptual and methodological foundations of the assessment tool, in particular please see the revised “Tool to analyze policies” subsection.

Specifically, we now describe (1) the literature-informed process used to identify and synthesise policy items from prior institutional COI governance frameworks; (2) the rationale for domain structure based on areas where conflicts of interest have been shown to influence medical society activities – for each domain we now include a reference; (3) the iterative expert review process involving 10 researchers with extensive experience in COI policy development and analysis; and (4) the rationale for the scoring system and equal weighting of items to ensure transparency and avoid subjective prioritisation. We also note that this development approach is consistent with established methods used in prior institutional COI policy assessment tools, specifically in academic medical centres.

Reviewer #2 comments

2. It is crucial to have a robust justification for the decision and its ramifications, as limiting the search to public websites represents a major methodological choice.

Authors’ response

We thank you for highlighting the need to more clearly justify this methodological decision. We have revised the Methods section to clarify that restricting the search to publicly accessible policies was intentional and to explain the implications of this choice. The revised text now states: “Policies were identified through a systematic search of publicly accessible sections of each society’s website.” We further clarify the rationale for this approach: “We focused on publicly available policies because these represent the standards communicated to external stakeholders and therefore constitute the most transparent and publicly accountable form of organizational governance.”

We also explain the implications for reproducibility and data collection, noting that “Restricting the search to publicly accessible materials also ensured methodological reproducibility, as access to password-protected or internal documents varies across organizations and investigators.” Finally, we clarify that societies were not contacted and specify the scope of the analysis: “We did not contact medical societies to request additional policies because our objective was to evaluate policies accessible to any interested party. Consequently, our analysis reflects the policies that societies make publicly available rather than any internal policies that may exist.”

Reviewer #2 comments

3. Tables 3, 4, and 6 are dense and somewhat confusing. Table 3's "Depth of coverage" columns are hard to parse. Consider simplifying. For Table 6, the "% of Max" column is useful but requires careful reading. Ensure the caption/footnote clearly explains that a lower percentage indicates a more restrictive overall policy stance.

Authors’ response

Thank you for your reflections on the Tables, this is very helpful and we have made some amendments as a result.

To address the confusing presentation of Table 3 (now Table 4), we have divided it into two separate tables. Previously, it was combining breadth and depth which we agree made it confusing for the reader. Splitting the Table into two also has the added benefit of better aligning with the text, as we have a section for breadth and a section for depth.

We have added an additional sentence to the footnote for Table 6 (now Table 7) to clarify how readers can interpret the scores – “*Percentage of maximum represents the absolute score as a percentage of the maximum (most permissive) possible score if all items were scored as 2. Lower percentages indicate more restrictive overall policy positions, whereas higher percentages indicate more permissive policies.”

Reviewer #2 comments

4. The low "depth" scores are highlighted. This could be interpreted not only as a policy weakness but also that the tool is very comprehensive. This dual interpretation should be acknowledged.

Authors’ response

We appreciate this point and have inserted an additional sentence to reflect the dual interpretation, as you suggested. We now state: “Even within policies, depth of coverage was modest: on average, societies addressed fewer than one-third of the relevant items within each domain. While this could be interpreted as a policy weakness, it may also reflect the comprehensiveness of our tool.”

Reviewer #2 comments

5. The discussion suggests policies are "reactive" and "compliance-driven." While plausible, this is still a speculative from a cross-sectional study. It is thus strongly suggested to refine informal phrasing (such as “implying” or “indicating”) to underscore the relational elements within your observations.

Authors’ response

Thank you for highlighting this. We appreciate and agree there is a need to avoid speculative interpretations and causal language as this is not supported by our research design.

We have revised with discussion section to use more cautious, relational phrasing that emphasises observed policy characteristics rather than institutional intent or motivations. Specifically:

- We replaced stronger interpretative language with observational phrases (e.g. "more frequently observed”, “more frequently included”, “consistent with”, “may suggest”, and “may have implications”).

- We reframed statements about policy orientation to focus on structural patterns (e.g. “consistent with COI governance being structured around meeting external requirements”) rather than implying institutional motives

- We clarified interpretative statements by introducing hedging language (e.g. “may”, “suggest”, “comparatively”).

- We revised statements about potential impacts to emphasise implications rather than direct causal effects.

We retained one instance of the term “reactive”, but softened the phrasing to: “This pattern may suggest a comparatively reactive rather than proactive approach,” which we believe conveys a cautious interpretation of observed policy features rather than a claim about organisational intent.”

We feel these revisions ensure that our discussion remains proportionate to the study design while preserving the policy-relevant insights of our findings.

Reviewer #2 comments

Minor comments

1. In the abstract, the authors wrote "Systematically searched their websites".

Could be slightly more specific (e.g., "publicly accessible sections of their websites").

Authors’ response

Thank you for this helpful suggestion. We have clarified the scope of our search by revising the sentence to specify that only publicly accessible website content was reviewed. The sentence now reads: “We identified 68 Canadian medical societies meeting our inclusion criteria and systematically searched the publicly accessible sections of their websites for COI policies.”

Reviewer #2 comments

2. Line 38: Typo: "a 38 nd staff COI" -> "and staff COI".

Authors’ response

Thank you for pointing this out. This typo did not appear on our original Word copy, but may have occurred during the submission process. We have ensured that it does not appear in the revised version.

Reviewer #2 comments

3. Line 288: "The analysis of the policies does not apply to policies of excluded Canadian organizations..." This sentence is awkward. Rephrase for clarity.

Authors’ response

Thank you for pointing this out to us. We have now rephrased this sentence so it reads: “Our findings do not extend to organizations excluded under our eligibility criteria (e.g. regulatory colleges, accrediting bodies, broad physician associations, or multidisciplinary organizations) or to medical societies in other countries.”

Reviewer #2 comments

4. In Discussion section (Subtle: International Comparison): The comparison with the Australian study (Kerridge et al. 2005) is excellent as it contrasts your finding on speaker declarations. Emphasize this point of difference more.

Authors’ response

Thank you for identifying this as an important point of contrast between our findings and other contexts. We have sharpened the phrasing to draw more attention to this fact by adding an additional sentence:

“In Australia, Kerridge et al. (19) reported that 20 of 29 responding organizations (from 63 contacted) had policies or guidelines governing industry relationships. However, these policies appeared limited in scope: 62% of organizations did not require speakers to declare COIs at educational meetings. This contrasts with our findings, where speaker declaration requirements was the most frequently addressed policy item across all items in our study. The increase in the percent of policies requiring speaker disclosure may reflect an expanding recognition of the influence of COI”

---

## [Decision Letter · Decision Letter 1]

14 Apr 2026

Evaluating conflict-of-interest governance among Canadian medical societies using a novel assessment tool: a cross-sectional study

PONE-D-25-63289R1

Dear Dr. Joel Lexchin,

We’re pleased to inform you that your manuscript has been judged scientifically suitable for publication and will be formally accepted for publication once it meets all outstanding technical requirements.

Kind regards,

Tope Michael Ipinnimo, MBBS, MPH, FWACP, FMCPH

Academic Editor

PLOS One

Additional Editor Comments (optional):

Reviewers' comments:

Reviewer's Responses to Questions

**Comments to the Author**

1. If the authors have adequately addressed your comments raised in a previous round of review and you feel that this manuscript is now acceptable for publication, you may indicate that here to bypass the “Comments to the Author” section, enter your conflict of interest statement in the “Confidential to Editor” section, and submit your "Accept" recommendation.

Reviewer #1: All comments have been addressed

Reviewer #2: All comments have been addressed

2. Is the manuscript technically sound, and do the data support the conclusions?

Reviewer #1: Yes

Reviewer #2: Yes

3. Has the statistical analysis been performed appropriately and rigorously?

Reviewer #1: Yes

Reviewer #2: Yes

4. Have the authors made all data underlying the findings in their manuscript fully available?

Reviewer #1: Yes

Reviewer #2: Yes

5. Is the manuscript presented in an intelligible fashion and written in standard English?

Reviewer #1: Yes

Reviewer #2: Yes

6. Review Comments to the Author

Reviewer #1: The authors have adequately addressed the concerns raised in the previous round of review. One grammatical error was introduced in the newly added text in the Discussion section (International Comparison): "speaker declaration requirements was the most frequently addressed policy item" - "requirements" is plural and should read "were" instead of "was." This should be corrected prior to publication.

Reviewer #2: All issues of concern have been meticulously addressed. The manuscript is now methodologically rigorous, clearly presented.

7. PLOS authors have the option to publish the peer review history of their article (what does this mean?). If published, this will include your full peer review and any attached files.

Reviewer #1: No

Reviewer #2: No

---

## [Editor Report · Acceptance letter]

PONE-D-25-63289R1

PLOS One

Dear Dr. Lexchin,

I'm pleased to inform you that your manuscript has been deemed suitable for publication in PLOS One. Congratulations! Your manuscript is now being handed over to our production team.

Kind regards,

on behalf of

Dr. Tope Michael Ipinnimo

Academic Editor

PLOS One